# Comparison of Multi-locus Genotypes Detected in *Aspergillus fumigatus* Isolated from COVID Associated Pulmonary Aspergillosis (CAPA) and from Other Clinical and Environmental Sources

**DOI:** 10.3390/jof9030298

**Published:** 2023-02-24

**Authors:** Susana Morais, Cristina Toscano, Helena Simões, Dina Carpinteiro, Carla Viegas, Cristina Veríssimo, Raquel Sabino

**Affiliations:** 1Faculdade de Ciências, Universidade de Lisboa, 1749-016 Lisboa, Portugal; 2Reference Unit for Parasitic and Fungal Infections, Department of Infectious Diseases, National Institute of Health, Av. Padre Cruz, 1649-016 Lisbon, Portugal; 3Microbiology Laboratory, Centro Hospitalar Lisboa Ocidental, Hospital Egas Moniz, 1349-019 Lisboa, Portugal; 4Technology and Innovation Unit, Department of Human Genetics, National Institute of Health, 1649-016 Lisbon, Portugal; 5H&TRC-Health & Technology Research Center, ESTeSL-Escola Superior de Tecnologia da Saúde, Instituto Politécnico de Lisboa, 1990-096 Lisbon, Portugal; 6Public Health Research Centre, NOVA National School of Public Health, Universidade NOVA de Lisboa, 1600-560 Lisbon, Portugal; 7NOVA National School of Public Health, Public Health Research Centre, Comprehensive Health Research Center (CHRC), 1600-560 Lisbon, Portugal; 8Instituto de Saúde Ambiental, Faculdade de Medicina, Universidade de Lisboa, 1649-028 Lisboa, Portugal

**Keywords:** *Aspergillus fumigatus*, CAPA, STR*Af* assay, microsatellite, genotyping

## Abstract

Background: *Aspergillus fumigatus* is a saprophytic fungus, ubiquitous in the environment and responsible for causing infections, some of them severe invasive infections. The high morbidity and mortality, together with the increasing burden of triazole-resistant isolates and the emergence of new risk groups, namely COVID-19 patients, have raised a crescent awareness of the need to better comprehend the dynamics of this fungus. The understanding of the epidemiology of this fungus, especially of CAPA isolates, allows a better understanding of the interactions of the fungus in the environment and the human body. Methods: In the present study, the M3 markers of the STRAf assay were used as a robust typing technique to understand the connection between CAPA isolates and isolates from different sources (environmental and clinical-human and animal). Results: Of 100 viable isolates that were analyzed, 85 genotypes were found, 77 of which were unique. Some isolates from different sources presented the same genotype. Microsatellite genotypes obtained from *A. fumigatus* isolates from COVID+ patients were all unique, not being found in any other isolates of the present study or even in other isolates deposited in a worldwide database; these same isolates were heterogeneously distributed among the other isolates. Conclusions: Isolates from CAPA patients revealed high heterogeneity of multi-locus genotypes. A genotype more commonly associated with COVID-19 infections does not appear to exist.

## 1. Introduction

*Aspergillus fumigatus* is an environmental saprophyte fungus with a ubiquitous distribution in the environment (air, water, soils) and organic matter [1]. This species has an increasing relevance in animal and especially in human health, causing a wide spectrum of diseases [2], and having a relatively high morbidity and mortality [3]. Recently, the World Health Organization (WHO) released the WHO fungal priority pathogens list (FPPL), the first-ever list of health-threatening fungi [4], and *A. fumigatus* was included in the critical priority group of fungi due to the impact on health and the problem of resistance emergence.

*Aspergillus fumigatus* conidia are able to enter the respiratory tract of both humans and animals, and in cases of host susceptibility, this species is capable of infecting the lower respiratory tract and causing chronic, allergic, or invasive disease [5,6,7]. Clinical risk factors usually associated with the occurrence of disease include immunosuppression, frequently associated with neutropenia in hematological and transplant patients for example, immunosuppressive treatments, and some other comorbidities [8]. In immunocompetent individuals, conidial inhalation usually leads simply to colonization (followed by removal through mechanical and immunological defenses), and in the majority of cases, there is no respiratory tract infection. However, when infected, immunocompetent individuals often present chronic or allergic forms of the disease, since their immune system is able to prevent an invasive infection [2,9]. Nevertheless, there are also cases of immunocompetent patients who develop invasive infections, especially when exposed to a high burden of conidia, mainly in workplaces where high amounts of organic matter are manipulated, representing an occupational hazard [10,11]. Some other patient groups appear to be associated with infections triggered by *Aspergillus*, namely ICU patients, especially those who require intubation [12], patients with influenza [13], and COVID-19 positive patients [14,15].

COVID-19 associated pulmonary aspergillosis rises as a new worrying entity, particularly relevant in this global COVID-19 pandemic. COVID-19 associated pulmonary aspergillosis patients do not exhibit the usual risk factors associated with aspergillosis, and, therefore, CAPA diagnosis poses new challenges [16]. Some guidelines have been published in order to establish standard criteria for CAPA diagnosis and treatment [17].

Due to the lack of understanding about COVID-19 at the beginning of the pandemic and the need to assure medical personnel safety, bronchoscopy was rarely performed in COVID positive patients [18]. Hence, few specimens of *Aspergillus* were recovered to allow for diagnosis of aspergillosis. To overcome this problem, some other respiratory samples such as tracheal aspirates, sputum, and bronchial secretions were therefore used for CAPA diagnosis [19,20].

It is still not well understood why viral infections, such as SARS-CoV-2 infections, pose a risk factor for *Aspergillus* infections. It is possible that when the viral infection occurs, there is an exacerbated inflammatory response by the host’s immune system, which leads to severe acute respiratory distress syndrome (ARDS). ARDS causes pulmonary damage, possibly responsible for the high risk of the development of secondary infections, including fungal infections. Adding to that, the use of some drugs for COVID-19 treatment, like corticosteroids, may also increase the susceptibility to secondary infections [21,22,23]. Taking this into account, there are no typical risk factors associated with CAPA, and there are several cases of CAPA in immunocompetent patients, without any prior comorbidity, that end up being fatal [24,25]. Adding up to the challenge that the disease represents, there is a rising problem with the emergence of triazole-resistant isolates, especially those with the TR34/L98H mutation on the *cyp*51A gene [15,16]. Isolates harboring this mutation show high resistance to several triazoles, which makes treatment of infections caused by these resistant strains even more difficult and results in poorer outcomes [17].

The high incidence of CAPA in some countries, its associated mortality [26,27], as well as the limitations in diagnosis, highlight the relevance of epidemiological studies on this issue, in order to reach a better understanding of fungal dynamics.

Genotyping is a very efficient approach to better understand the molecular epidemiology of *A. fumigatus* because it allows the distinction among different strains of a species. The STR*Af* (Short tandem repeats of *Aspergillus fumigatus*) assay allows the amplification of polymorphic loci within the fungal genome in a multiplex reaction and allows for the determination of the number of repeats for each locus. It is a very robust, reproducible method [28,29]. It is useful to better understand outbreaks, to have a better perception of geographical or timely strain distribution, and to determine if there are genotypes more frequently associated with infection. Hence, this microsatellite-based typing method using three loci composed of tandemly repetitive stretches of three nucleotides was applied to genotype *A. fumigatus* sensu stricto isolates. Our aim is to study *A. fumigatus* isolates from CAPA patients and to compare them with isolates from different origins by application of these microsatellite markers to understand how our isolates are related to *A. fumigatus* isolates from different sources and geographical distributions. Our goal is to distinguish epidemiologically related isolates, identify prevalent strains and genotypes associated with this disease, compare environmental and clinical isolates, and determine possible routes of acquisition of a strain or to identify possible reservoirs.

## 2. Materials and Methods

### 2.1. Selection of Aspergillus Isolates 

Eleven *Aspergillus fumigatus* isolates obtained from respiratory samples of 10 patients diagnosed with COVID-19 were selected for this study (characterized in Table 1). In parallel, 95 more *A. fumigatus* isolates were added to the study for comparison (45 environmental, 38 from respiratory samples from non-COVID+ patients and 6 from deep-seated infections also of non-COVID+ patients, and 6 from clinical animal sources (isolates characterized in Appendix A). These isolates were randomly selected from a database of more than 400 *Aspergillus* isolates available at the Mycology National Reference Laboratory. The isolates were cultured onto malt extract agar medium supplemented with chloramphenicol, and DNA was extracted from a saline solution of spores, using the High Pure PCR Template Preparation kit (Roche Diagnostics Corp., Indianapolis, IN, USA) according to the manufacturer’s instructions.

### 2.2. Molecular Identification of Isolates

To identify the selected isolates at the species level, partial sequencing of the calmodulin gene was carried out. Briefly, a polymerase chain reaction (PCR) was performed in a 25 µL volume using Cytiva PureTaq Read-to-Go beads (Life Sciences IP Holdings Corporation Washington, DC, USA) with 0.7 µM of the cmd5 (5′-CCG-AGT-ACA-AGG-AGG-CCT-TC-3′) and cmd6 (5′-CCG-ATA-GAG-GTC-ATA-ACG-TGG-3′) primers [30]. The PCR was carried out in a thermocycler under the following conditions: an initial denaturation step of 95°C for 10 min followed by 38 cycles at 95 °C for 30 s, 55 °C for 30 s, and 72 °C for 1 min. A final extension was performed at 72 °C for 1 min. PCR products were analyzed by 2% agarose gel electrophoresis and purified using the IlustraTM ExoStar enzyme system (GE Healthcare, Chicago, IL, USA). 

Sequencing of the forward chain of the gene was performed using the BigDye terminator sequencing kit with the cmd5 primer. The conditions of this PCR were as follows: an initial denaturation at 96 °C for 10 s, 30 cycles at 96 °C for 30 s, 50 °C for 5 s, and 60 °C for 4 min, followed by a final extension at 72 °C for 7 min.

The samples were then subjected to Sanger sequencing in the 3500 Genetic Analyzer (Applied Biosystems^TM^, Waltham, MA, USA). The obtained sequences were edited in Chromas 2.6.6 software (Technelysium Pty Ltd., South Brisbane, Australia), and species identification was obtained by comparison with the sequences deposited in the BLAST platform [31]. A minimum homology of 98% was the requirement to consider an acceptable identification to species level. 

### 2.3. Microsatellite Genotyping 

The *A. fumigatus* sensu stricto isolates characterized in Appendix A were subjected to genotyping using the STR*Af* assay previously described by de Valk et al. [32], with a high discriminatory power (0.9994). Given the high discriminatory power of each of the three multiplex reactions of the *STRAf* assay [32] only the M3 markers, a panel of trinucleotide markers, were used in this study. Briefly, three sets of primers were used to amplify the three selected loci–STR*Af* 3A labeled with FAM (carboxyfluorescein), STR*Af* 3B labeled with HEX (hexaclorocarboxyfluorescein), and STR*Af* 3C labeled with NED (Table 2).

PCR conditions were previously described [32]. Following amplification, PCR products were subjected to fragment analysis. Briefly, each PCR product was diluted in distilled water (1:30), and 1 µL of the diluted samples was then combined with the GeneScan™ 500 ROX™ marker (Applied Biosystems™,Waltham, MA, USA). The mixture was subjected to thermal denaturation (3 min at 95 °C followed by a quick cooling to 4 °C). The treated samples were subjected to capillary electrophoresis in the 3500 Genetic Analyzer (Applied Biosystems^TM^ Waltham, MA, USA).

The strain CD10 was used in every performed instance of capillary electrophoresis, as positive and reproducibility control. A sample with water in the place of DNA was used in every run as a negative control.

### 2.4. Fragment Analysis and Genetic Relationships

The fragments corresponding to the amplification of each locus were analyzed in the GeneMapper™ Software 6 (Applied Biosystems^TM^, Waltham, MA, USA) to determine their size in base pairs (bp). This length was further converted into the number of repeats, using calculations and reference strains.

The alleles obtained for each isolate were then analyzed in the Bionumerics 6.6 software (Applied Maths, bioMérieux SA, Marcy-l’Étoile, France) using a categorical similarity coefficient and an unweighted paired group method with arithmetic mean (UPGMA) for the cluster analysis. The minimum spanning tree (MST) and dendrogram were obtained using this software.

### 2.5. Genetic Relations with AfumID Isolates

The number of repeats for each allele from every isolate was included in the AfumID database (https://afumid.shinyapps.io/afumID/ (accessed on 21 June 2022), and compared with the 4049 isolates from the database [33].

## 3. Results

### 3.1. Allele Characterization

After being submitted to the species identification as described in point 2.2 of the material and methods section, DNA of the selected *A. fumigatus* sensu stricto isolates (N = 106) was used for the genotyping assay. The electropherograms obtained for each isolate were analyzed, and the fragment size was determined for the three loci. The number of repeats of each isolate was calculated, according to what is described in point 2.4. Overall, microsatellite multi-locus genotypes of 100 viable isolates were considered. The analysis of these 100 isolates showed that all the microsatellite loci were polymorphic, presenting between 17 and 37 alleles. The size ranges of the alleles as well as the corresponding number of repeats are shown in Table 3. From the three loci analyzed, STRAf 3A had the highest number of different alleles, and also the widest range of allele sizes. On the other hand, STRAf 3B had the smallest range and the lowest number of different alleles.

From the referred 100 isolates, 85 different multi-locus genotypes were obtained. Of them, 90.6% (77 out of 85) were unique and only 9.4% (8 out of 85) represented genotypes that appeared more than once.

### 3.2. Genetic Relationships

The minimum spanning tree (MST) of the studied isolates shows that there is not an obvious grouping of isolates by source and that the isolates with the same genotype are mostly from environmental sources (the bigger light green circles) (Figure 1). Isolates with known resistance to triazoles (any source), were highlighted using a different color (pink), in order to understand if there was any clustering of the resistant isolates, which was not observed (Figure 1). In the MST, there is no clustering of the isolates obtained from COVID-19+ patients (red circles), and they are heterogeneously spread across the tree. None of the genotypes of *A. fumigatus* isolates collected from COVID+ patients is common to more than one isolate; they were all unique (Figure 1).

A dendrogram was also constructed (Appendix A), and the most relevant sections with the closest genetically related isolates are highlighted in Table 3. All the other studied isolates (not shown in Figure 2) presented a unique multilocus genotype. 

Isolates collected from COVID-19+ patients are heterogeneously scattered through the dendrogram. Each one of these isolates presents a unique microsatellite multi-locus genotype, as referred to above. Yet, in cluster #4, isolates VA388 and VA394 are closely related, presenting a difference of only one repeat unit in one of the loci. Isolate VA382, also from cluster #4, shares only one allele with VA388 and VA394 isolates, which are genetically closer to CD257, an environmental isolate (sharing two alleles). Cluster #14 includes isolate VA392, which shares two alleles (STR*Af* 3A and 3B) with a non-COVID respiratory product, VA378, which shows a difference of one repeat unit in STR*Af* 3C. Isolates VA390 and VA394 were collected from the same patient (two days apart). These isolates did not cluster together, and their multi-locus genotypes were completely different. This can represent a mixed infection with two different strains at the same time, a fact that is known to happen with *A. fumigatus* infections [34]. 

In other groups that do not include *A. fumigatus* isolates from patients positive for COVID-19, it is possible to perceive that some clusters are formed by isolates with exactly the same microsatellite multi-locus genotype: clusters #1 and #6 comprise four environmental isolates each with the same genotype, and cluster #9 comprises four isolates from respiratory samples also presenting the same genotype. Some other clusters, namely #2, #3, #7, #11, and #16 also include genotypes that are not unique and are shared between two or three isolates. The remaining isolates presented in Figure 2 all showed a unique genotype, even though some of them share one or two alleles with one or more isolates (Appendix A). Cluster #10 includes two isolates collected from animals that are closely related and differ only in one allele.

### 3.3. AfumID 

The isolates enrolled in our study were included in the database AfumID (https://afumid.shinyapps.io/afumID/ (accessed on 21 June 2022). Figure 3 shows how the COVID+ isolates take place among the ones already inserted in the database (A), and how the set of all isolates of our study are scattered throughout the database (B). The orange dots correspond to the susceptible isolates from the database, the blue dots represent the resistant ones, and the black dots represent the isolates inserted by us in the database.

## 4. Discussion

The rising relevance of *Aspergillus fumigatus* in human health has brought an increasing interest in its epidemiology. The STR*Af* assay is a very useful tool to conduct genotyping studies due to its specificity, robustness, and high discriminatory power [32]. Isolates collected from COVID positive patients, together with isolates from different sources, were analyzed as described, and the genetic relationships between them were analyzed. It was possible to observe a heterogeneous distribution of isolates among the different sources. However, some of the isolates share one or two alleles with other isolates.

Analyzing the clusters highlighted in Figure 2, it is possible to conclude that those containing isolates with exactly the same multi-locus genotype are mostly formed by environmental isolates (clusters #1, #2, #3, and #6). However, cluster #9 is composed of four respiratory samples presenting the same genotype. In contrast to what happens in studies like the one performed by Guinea et al. [35] where environmental isolates usually exhibit higher genetic diversity than respiratory ones, our environmental isolates presented less diversity. This may be due to the fact that some of the environmental isolates were recovered in the same spatial location or indoor environment, which might lead to a bias in the obtained data. 

When dealing with an environmental fungus capable of infecting both humans and animals, and with a rising problem of resistance of environmental origin [36,37], it is also important to better understand how isolates from these sources relate to each other. One relevant observation in this study is the importance of *Aspergillus fumigatus* in the One Health context. In the present study, isolates from different sources appeared to be closely related, some of them with exactly the same genotypes. This is the case with isolates HSMA34 and VA105 (cluster #7), an environmental and respiratory isolate, respectively, isolates VA225 and VA290 (cluster #11) a human respiratory isolate, and an animal one and isolates VA85 and VA95 (cluster #16), two isolates with known azole resistance, with a respiratory and environmental origin, respectively. This observation will be crucial for the implementation of measures to avoid the exposure of patients in healthcare facilities and workers or occupants in other indoor settings.

Every isolate collected from a COVID+ patient presented a unique genotype. Some of these isolates showed higher phylogenetic proximity to each other, more than with other isolates collected from COVID negative patients. Cluster #4 includes two COVID+ isolates with only one repeat unit of difference in STR*Af* 3A locus–this might represent the occurrence of a microvariation event, since both isolates were recovered in the same hospital, but separated by months, and the detected variation occurs only in one marker [38]. VA382, also from the same cluster, is more closely related to isolates collected from non-COVID+ patients, namely with an environmental isolate (CD257) and with an isolate collected from a human respiratory specimen (VA221), sharing 2 alleles with each of them. Another cluster with closely related isolates includes isolates collected from respiratory products from a COVID+ patient (VA392) and isolates from a non-COVID patient (VA378). Both isolates were collected from patients hospitalized in the same hospital and at the same time. This cluster (#14), gathers two isolates that share two alleles and have a difference of one repeat unit in the remaining allele, which might be a case of microvariation. Curiously, these isolates (VA378 and VA392) have an especially high number of repeats in locus STR*Af* 3B. A recent study by Steenwyk et al. [39] reveals that there are no significant differences in the genome of COVID+ isolates and a reference strain used in that study. In our study, all the other isolates collected from COVID+ patients were closely related to isolates from other sources, sharing one or two alleles with them. The heterogeneity of genotypes displayed supports the data presented previously by Mead [40]. Interestingly, two COVID+ isolates (VA390 and VA394) were collected from the same patient but displayed totally different multi-locus genotypes, which suggests that the patient was colonized at least by two different strains, as previously referred to by other authors [34].

Even though none of the COVID+ isolates have the same multi-locus genotype in this study, it is not surprising that some isolates present close genetic relationships, because almost all were obtained from patients in the same hospital, and therefore the patients might have been exposed to the same contamination source. The study of Peláez-García et al. [41] however, concluded that *Aspergillus* infections in COVID-19+ patients seem to be community acquired, and not of nosocomial origin.

The inclusion of our isolates in the AfumID database allows us to understand the placement of our isolates in the general population of the isolates that are included in that database (Figure 3). This database is composed of the multi-locus genotypes of 4049 isolates collected all over the world, from environmental and clinical sources. The database is composed of isolates susceptible to azoles, and others with known azole resistance associated with the TR34/L98H and TR46/ Y121F/T289A mutations in the *cyp51A* gene. Most of the resistant isolates are grouped in one clade of the database, forming the resistant clade. It is possible to observe that most of the isolates of our study are placed in the susceptible clade, but some of them are placed in the resistant clade. It would be expected that isolates with known resistance (Appendix A) caused by the TR34/L98H mutation would be grouped in this clade, which was observed. Resistant isolates bearing other mutations were not all grouped in this clade, which could be explained by the fact that this database was built with susceptible isolates and with TR34/L98H resistant isolates [33]. The majority of the isolates without detected azole resistance were grouped, as expected, in the susceptible clade, but some of them were grouped in the resistant clade, which may be due to the presence of resistant isolates with similar multi-locus genotypes. 

Our study presented some limitations: almost all of the isolates collected from COVID+ patients were from the same hospital and this could represent a bias in our results. However, since none of the isolates presented the same multi-locus genotype, this situation may not represent a problem. The heterogeneity of genotypes among isolates collected from COVID+ patients was also supported by Peláez-García et al. [41]. The small number of isolates collected from COVID+ patients due to the lack of performed bronchoscopies [18] is also a limitation that we could not overcome, since not all hospitals send their samples to our laboratory.

## 5. Conclusions

To our knowledge, this was the first Portuguese study on *Aspergillus fumigatus* molecular epidemiology that includes isolates retrieved from COVID-19 positive patients. Even with a limited sample size, it was possible to perceive the heterogeneity of these isolates through the determination of their microsatellite multi-locus genotypes, and a genotype more frequently associated with SARS-CoV-2 infection does not appear to exist. This study allowed us to understand more about the genetic relationships and position of isolates collected from COVID+ patients in the background of other isolates collected in our country over the years.

The possibility to perform multicentric studies with *Aspergillus* isolates from COVID+ patients from different hospitals throughout the country and to include other isolates from more diverse sources will allow for the possibility to better understand *Aspergillus fumigatus* interaction with SARS-CoV-2 infection and better manage the rising number of aspergillosis cases, especially the ones linked with viral infections. Additionally, this will allow sharing of preventive measures in different indoor environments to avoid exposure to *Aspergillus* contamination.

## Figures and Tables

**Figure 1 jof-09-00298-f001:**
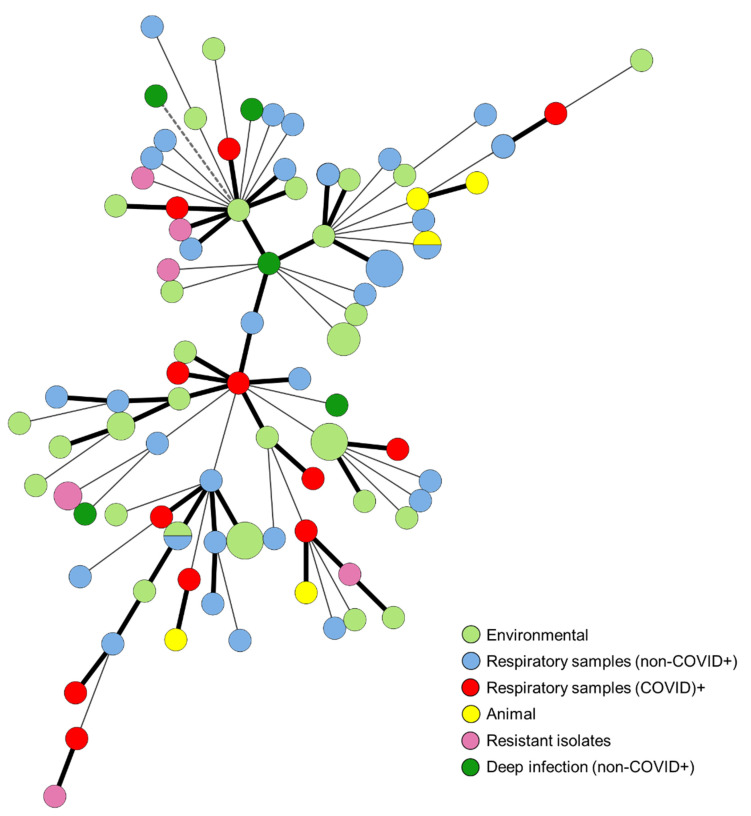
Minimum spanning tree constructed in Bionumerics 6.6. software (Applied Maths, bioMérieux SA, Marcy-l’Étoile, France) using a categorical similarity coefficient and an unweighted pair group method with arithmetic mean (UPGMA) analysis. The 85 different genotypes found are represented by each circle. The circle size is proportional to the number of isolates with each genotype. Each color represents a different isolate source, as shown in the bottom left corner. Each line represents the distance between genotypes: solid bold line–1 marker; solid line–2 markers, and dotted line–3 markers.

**Figure 2 jof-09-00298-f002:**
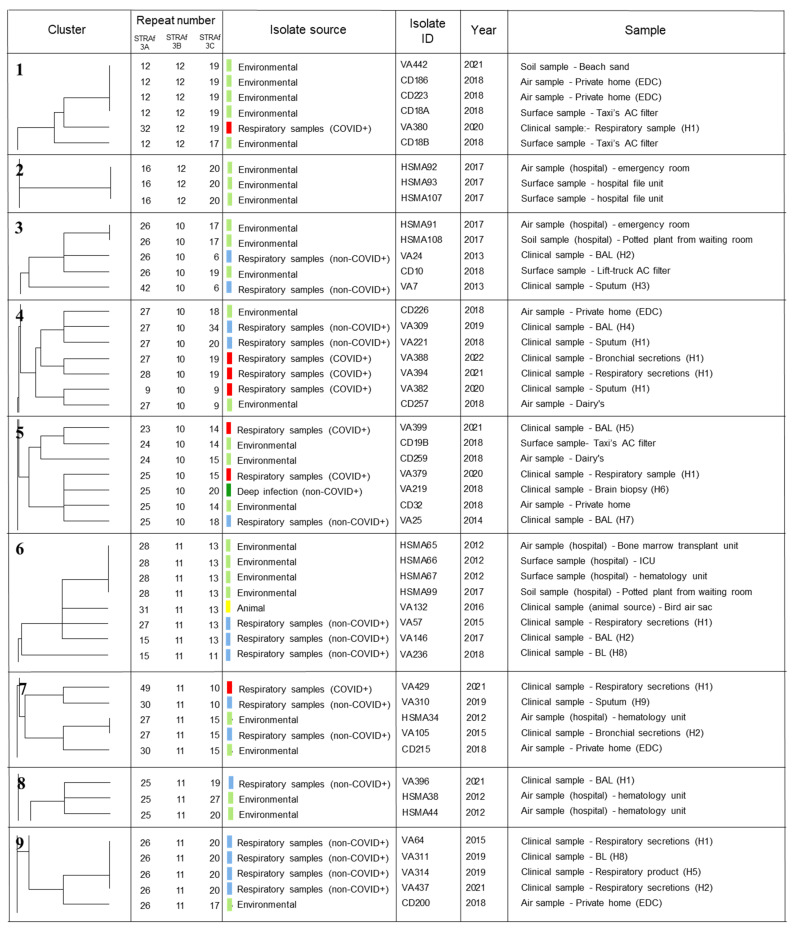
Groups of isolates showing genetic relatedness obtained from the dendrogram constructed in Bionumerics 6.6. software (Applied Maths, bioMérieux SA, Marcy-l’Étoile, France) using a categorical similarity coefficient and an unweighted pair group method with arithmetic mean (UPGMA) analysis. AC–air conditioner; EDC–electrostatic dust collector; BAL–bronchoalveolar lavage; BL–bronchial lavage; H1 to H10 –hospitals’ ID (from where the isolates were collected); R–resistant to azoles.

**Figure 3 jof-09-00298-f003:**
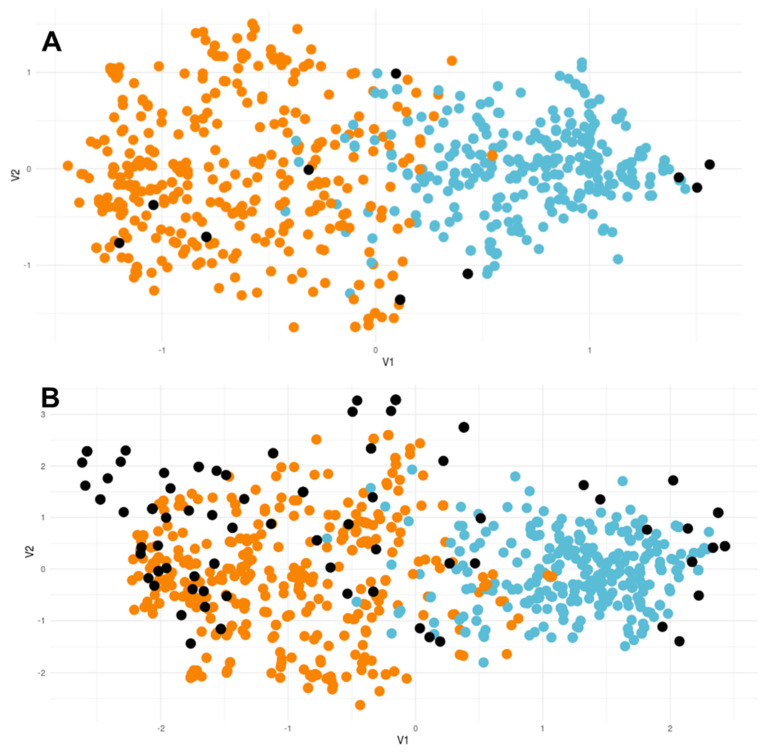
Representation of the isolates in this study (black dots) in the general population of the AfumID database. The orange clade represents susceptible isolates. The blue clade represents resistant isolates. (**A**). Set of isolates collected from COVID+ patients. (**B**). Set of all isolates in this study.

**Table 1 jof-09-00298-t001:** Characterization of the isolates collected from COVID-19 patients.

Isolate Identification	VA 379	VA 380	VA381	VA 382	VA 388	VA 390 VA 394	VA 392	VA399	VA 429	VA 443
**Date of isolation**	10 December 2020	10 December 2020	12 November 2020	9 November 2020	16 December 2021	20 January 2021 22 January 2021	14 December 2020	8 March 2021	25 July 2021	28 August 2021
**Demographics**										
Gender	Male	Male	Male	Male	Female	Male	Female	Male	Male	Male
Age (y)	66	78	49	81	74	69	97	64	72	34
**Underlying conditions**										
Diseases	COPD (steroids) hypertension, hypothyroidism, ex-smoker (for >10 y)	COPD (steroids), Alzheimer’s disease, osteoporosis, ex-smoker (for 17 y; previously > 40 PY)	None	Smoker	Hypertension,	COPD (steroids), obesity, hypertension, dyslipidemia,	Hypertension	INA	Hypertension,	None
				COPD	Hipotiroidism	insulin-dependent diabetes,			Diabetes mellitus, Obesity,	
				Cardiac insufficiency	Diabetes mellitus, Obesity	1st degree AV block, psoriasis,			Dislipidemia,	
				Acute renal lesion		vitiligo, previous hepatitis B, ex-smoker (for > 35 y)			Colon adenocarcinoma Pancitopeniae	
**ARDS**										
Mechanical ventilation	Yes	Yes	Yes	Yes	Yes	Yes	No	No	Yes	Yes
**Microbiology**										
Fungal culture	TA:	TA:	BALF:	TA:	TA:	TA:	TA:	BALF:	TA:	TA:
	*Aspergillus* section *Fumigati*	*Aspergillus* section *Fumigati*	*Aspergillus* section *Fumigati*	*Aspergillus* section *Fumigati*	*Aspergillus* section *Fumigati*	*Aspergillus* section *Fumigati*	*Aspergillus* section *Fumigati*	Aspergillus section *Fumigati*	*Aspergillus* section *Fumigati*	*Aspergillus* section *Fumigati*
Molecular identification	*Aspergillus fumigatus sensu stricto*	*Aspergillus fumigatus sensu stricto*	*Aspergillus fumigatus sensu stricto*	*Aspergillus fumigatus sensu stricto*	*Aspergillus fumigatus sensu stricto*	*Aspergillus fumigatus sensu stricto*	*Aspergillus fumigatus sensu stricto*	Aspergillus fumigatus sensu stricto	*Aspergillus fumigatus sensu stricto*	*Aspergillus fumigatus sensu stricto*
BALF GM (≥1)	Not available	Not available	Negative	Not available	Not available	Not available	Not available	Positive	Not available	Not available
Serum GM (≥0.5)	Not available	Negative	Not available	Not available	Not available	Not available	Not available	Not performed	Not available	Not available
**Classification**										
AspICU (modified) Algorithm	Colonization	Colonization	Putative	Putative	Putative	Colonization	Colonization	Putative	Putative	Colonization
ECMM/ISHAM CAPA Criteria	Possible	Possible	Probable	Probable	Probable	Possible	Possible	Possible	Probable	Possible
**Therapy for *Aspergillus***										
Antifungal	Voriconazole	Voriconazole	Voriconazole	Voriconazole	Voriconazole	Voriconazole	None	INA	Voriconazole	None

Legend: ARDS = acute respiratory distress syndrome; BALF = bronchoalveolar lavage fluid; BW = body weight; COPD = chronic obstructive pulmonary disease; TA = tracheal aspirate; INA = information not available.

**Table 2 jof-09-00298-t002:** Primers’ sequences, fluorophores used, and respective repeat units amplified (based on [32]).

Primer	Sequence (5′→3′)	Repeat Unit
STR*Af* 3A	F: FAM-GCTTCGTAGAGCGGAATCAC	TCT
	R: GTACCGCTGCAAAGGACAGT	
STR*Af* 3B	F: HEX-CAACTTGGTGTCAGCGAAGA	AAG
	R: GAGGTACCACAACACAGCACA	
STR*Af* 3C	F: NED-GGTTACATGGCTTGGAGCAT	TAG
	R: GTACACAAAGGGTGGGATGG	

Legend: F–forward; R–reverse.

**Table 3 jof-09-00298-t003:** Characterization of the alleles obtained in the microsatellite genotyping study of 100 viable *Aspergillus fumigatus* sensu stricto isolates.

Microsatellite	Number of Alleles	Lower Size Allele (bp)	Higher Size Allele (bp)	Number of Microsatellite Repeats (Range)	Most Common Allele (Number of Isolates)
*STRAf* 3A	37	129.5	283.22	7–57	25 (9)
STR*Af* 3B	17	159.27	273.27	9–47	11 (29)
STR*Af* 3C	31	73.89	207.84	4–47	20 (14)

Legend: bp–base pairs.

## Data Availability

Data presented in this study is not publicly available due to privacy restrictions.

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
