# Peer review of "Comparison of Multi-locus Genotypes Detected in Aspergillus fumigatus Isolated from COVID Associated Pulmonary Aspergillosis (CAPA) and from Other Clinical and Environmental Sources"

_jof, 2023, doi:10.3390/jof9030298_

Round 1

Reviewer 1 Report

Regarding to the Aspergillus isolates in special for those that were obtained from COVID patients it will be necesarry to specify which were the clinical conditions of the patient (as if those patients met any CAPA criteria, clinical characteristics and  status of those patients, age,  any possible comorbidity an also the treatment they were receving (steroids, under mechanical ventilation or not, etc) and not only mention them as respiratory samples, so one can have a better idea about the pathogenicity of those isolates and not merely represent another colonizer. It also be of great interest to know the specific time of the year the samples were collected. 

Author Response

We thank the reviewer for the constructive comments, which we believe have improved the quality of this manuscript. Our response is in attahment.

Reviewer 2 Report

Dear authors

thank you very much for this article. 

undoubtedly represents a great contribution to medical mycology

kind regards

Author Response

We thank the reviewer for the comments to this manuscript. Our response is in attachment.
